# Instance-based Generalization in Reinforcement Learning

**Martin Bertran**†*
Electrical and Computer Engineering
Duke University
martin.bertran@duke.edu

**Natalia Martinez**†
Electrical and Computer Engineering
Duke University
natalia.martinez@duke.edu

**Mariano Phielipp**
AI Labs
Intel Corporation
mariano.j.phielipp@intel.com

**Guillermo Sapiro**
Electrical and Computer Engineering
Duke University
guillermo.sapiro@duke.edu

## Abstract

Agents trained via deep reinforcement learning (RL) routinely fail to generalize
to unseen environments, even when these share the same underlying dynamics
as the training levels. Understanding the generalization properties of RL is one
of the challenges of modern machine learning. Towards this goal, we analyze
policy learning in the context of Partially Observable Markov Decision Processes
(POMDPs) and formalize the dynamics of training levels as *instances*. We prove
that, independently of the exploration strategy, reusing instances introduces signifi-
cant changes on the effective Markov dynamics the agent observes during training.
Maximizing expected rewards impacts the learned belief state of the agent by
inducing undesired instance-specific speed-running policies instead of generaliz-
able ones, which are sub-optimal on the training set. We provide generalization
bounds to the value gap in train and test environments based on the number of
training instances, and use insights based on these to improve performance on
unseen levels. We propose training a shared belief representation over an ensemble
of specialized policies, from which we compute a consensus policy that is used
for data collection, disallowing instance-specific exploitation. We experimentally
validate our theory, observations, and the proposed computational solution over the
CoinRun benchmark.

## 1   Introduction

Deep Reinforcement Learning (RL) has enjoyed great success on a range of challenging tasks such
as Atari [1, 2, 3, 4], Go [5, 6], Chess [7], and even on real-time competitive settings such as online
games [8, 9]. A common model-free RL scenario consists of training an agent to interact with an
environment whose dynamics are unknown with the objective of maximizing the expected reward. In
many scenarios [10, 11, 12, 13, 14, 15] the agent learns its behaviour policy by interacting with a
finite number of game levels (instances), and is expected to generalize well to new, unseen levels
sampled from the same model dynamics. This setting presents generalization challenges that share
similarities with those of goal-oriented RL, where the agent is expected to generalize to new, related
goals [16]. It has been observed that generalization gap between training levels and unseen levels
can be significant [12, 17, 18, 19], especially when the environment has hidden information, such

---

as the case of Partially Observable Markov Decision Processes (POMDP) [20, 21, 22, 23, 24]. In this scenario, the agent only has access to observations instead of the full state of the environment; learning the optimal policy implicitly requires an estimation of the unseen state of the system at each time step, which is usually accomplished through an intermediary belief variable. Obtaining policies that generalizes well to new situations is a major open challenge in modern RL.

Here we provide a formulation to describe the generalization of agents trained on a finite number of levels sampled from an underlying POMDP, like in CoinRun [12]. We formalize training levels as *instances*; these are functions sampled from an underlying Markov process that deterministically map a sequence of actions into a sequence of states, observations, and rewards. These level-specific functions mirror the role of samples in traditional supervised learning, and can provide insight on memorization of training dynamics, a form of overfitting present in model-free RL that is distinct from differences in the observation process [18, 19] and lack of exploration [16]. We prove that training over a finite set of instances changes the effective environment dynamics in such a way that the basic Markov property of the underlying model is lost. This phenomenon encourages an agent to encode information needed to identify a training instance and its specific optimal policy, instead of the intended level-agnostic optimal policy (which is optimal on the original POMDP dynamics). We can intuitively relate this to the difference between speed-running a level [25], where a human player has perfect memory of the environment dynamics, upcoming hazards, and the sequence of optimal inputs to reach the intended target with minimal delay; and the conservative strategy a player adopts when faced with a new level that only leverages environment dynamics, and is able to reach the goal even without knowledge of upcoming hazards. We identify two sources of information that can be used to identify the level, the instance-dependent observation dynamics (e.g., background theme in CoinRun), and the instance dynamics themselves (e.g., the $i$-th level is the only one in the training set that has two enemies before the first platform). The latter issue is challenging to address in the POMDP scenario, since past observations are needed to infer the true state of the environment, but can also be misused to infer the specific instance the agent is acting on.

Based on these challenges, we make the following main contributions to the theory and computational aspects of generalization in RL:

- We formalize the concept of training levels as *instances*, which are defined by deterministic mappings from actions to states, observations, and rewards sampled from the underlying Markov process of the environment. We show that this instance-based view is fully consistent with the standard POMDP formulation. Instances provide an exploration-agnostic abstraction of what is the optimal learnable policy on a finite set of levels when the goal is to maximize expected discounted rewards on unseen levels.

- We prove that learning from instances changes the underlying game dynamics in such a way that the optimal belief representation learned by the agent may fail to capture the true, generalizable dynamics of the environment; inducing policies that are optimal on the observed instance set, but fail to generalize to the overall dynamics, this is corroborated empirically. The instance-based formulation allows us to relate the generalization error of the policy value with the information the agent captures on the training instances using tools from generalization in supervised learning [26, 27, 28]. Our analysis provides theoretical backing to previous empirical observations in [12, 17, 18, 19, 29].

- As a step towards solving the issue of non-generalizeable policies, and making the learned policies instance-independent, we leverage the formulation of Bayesian multiple task sampling [30] and propose using ensembles of policies that, while sharing a common intermediate representation, are specialized to instance subsets and then consolidated into a consensus policy. The latter is used to collect experience and discourage instance-specific exploits, and is also used on unseen environments. We provide experimental validation of our observations and proposed algorithmic approach using the CoinRun benchmark. Code available at github.com/MartinBertran/InstanceAgnosticPolicyEnsembles

## 2 Related work

The exact mechanism by which neural networks are able to learn a policy that generalizes despite being able to fully memorize their training levels is an open research area [31, 32]. Many of the recent RL works [16, 17, 18, 19] are based in the Markov decision process (MDP) assumption, where

the current observation contains sufficient information about the underlying state; this alleviates the memorization problem described here, but is insufficient for many scenarios which can otherwise be tackled via POMDPs [33, 34, 35, 36]. Some strategies to improve generalization have addressed observational overfitting, for example, by adding noise to the observations as in [12], where Cutout augmentation [37] is applied, or by randomizing the features of the visual component of the agent network [18]. The use of Information Bottlenecks [38, 39] to both reduce the amount of information captured from the observation and improve exploration has been advanced in [16, 17]. We combine the results in [26, 27, 28, 30] with the proposed instance-based learning paradigm to design a training scheme that reduces the generalization gap and improves the performance on unseen levels.

## 3   Preliminaries

We consider a POMDP formulation, where an environment is defined by a set of unobserved states $s \in \mathcal{S}$, rewards $r \in \mathbb{R}$, possible actions $a \in \mathcal{A}$, observation modalities $k \in \mathcal{K}$, and observations $o \in \mathcal{O}$ that satisfy the following transition ($T$,[2],) and observation ($O$) distributions in time $t$:

$$
\begin{aligned}
s_0 &\sim \mu, \quad o_0 \sim O(o|s_0, k), k \sim U_{|\mathcal{K}|}, \\
r_{t+1}, s_{t+1} &\sim T(r, s \mid a_t, s_t), \quad o_{t+1} \sim O(o \mid a_t, s_{t+1}, k).
\end{aligned}
\tag{1}
$$

Here $s_0$ and $o_0$ are the initial state and corresponding observation, $\mu$ is the initial state distribution. Given a discount factor $\gamma \in [0, 1]$ and access to states $s$, the goal of an agent is to find the policy $\pi : \mathcal{S} \to \Delta^{\mathcal{A}}$ that maximizes the value function or expected return for every state:

$$
\pi^*(a \mid s) = \arg\max_{\pi} V_{\pi}(s) = \arg\max_{\pi} \mathbb{E}_{\pi}[\textstyle\sum_{i=1}^{T} \gamma^{i-1} r_{t+i} | s_t = s], \forall s \in \mathcal{S}.
\tag{2}
$$

However, since states $s$ are not observable, an actionable policy can only depend on the past trajectories of observations, rewards, and actions. Throughout the rest of the text, we note $H_t = \{a_{0:t-1}, s_{0:t}, o_{0:t}, r_{1:t}\}$ as the full $t$-step history of actions, states, observations, and rewards; we note $H_t^o = \{a_{0:t-1}, o_{0:t}, r_{1:t}\}$ to indicate that states are omitted. Under these conditions the objective is

$$
\pi^*(a \mid H_t^o) \quad = \arg\max_{\pi} V_{\pi}(H_t^o) = \arg\max_{\pi} \mathop{\mathbb{E}}_{\substack{a_t \sim \pi(\cdot|H_t^o) \\ r_{t+1}, o_{t+1} \sim p(\cdot|H_t^o, a_t)}} [r_{t+1} + \gamma V_{\pi}(H_{t+1}^o)], \forall H_t^o.
\tag{3}
$$

Note that $p(r_{t+1}, o_{t+1}|H_t^o, a_t)$ (present in the value estimation) can be written as

$$
p(r_{t+1}, o_{t+1}|H_t^o, a_t) \quad = \sum_{s_t, s_{t+1}} O(o_{t+1}|a_t, s_{t+1}) T(r_{t+1}, s_{t+1}|a_t, s_t) p(s_t|H_t^o).
\tag{4}
$$

Here the distributions $O(o_{t+1}|a_t, s_{t+1})$ and $T(r_{t+1}, s_{t+1}|a_t, s_t)$ are intrinsic to the the environment, while $p(s_t|H_t^o)$ characterizes the uncertainty on the true state of the environment, and is implicitly estimated in the process of improving the policy. The following remark formalizes the notion that $H_t^o$ is only useful insofar as it allows the estimation of the underlying state $s_t$, and motivates the introduction of the belief; a result that is well known in the literature [21, 22, 23, 24]. Meaning that there is an optimal policy that outputs the same action distributions for two observable histories with the same hidden state distribution $p(s_t \mid H_t^o)$; any encoding (belief) of the observed history that is unambiguous w.r.t. $p(s_t \mid H_t^o)$ can also be used.

**Remark 1. [23] Belief states.** Given a POMDP and actionable policies $\pi(a \mid H_t^o)$; for any two observable histories $H_{t'}^o$, $H_{t''}^o$ such that $p(s_t \mid H_{t'}^o) = p(s_t \mid H_{t''}^o)$ we have

$$
\begin{aligned}
\max_{\pi} V_{\pi}(H_{t'}^o) \quad &= \max_{\pi} V_{\pi}(H_{t''}^o), \\
\exists \pi^*(a \mid p(s \mid H_t^o)) \quad &\text{such that } \pi^* \in \arg\max V_{\pi}(H_t^o), \forall H_t^o.
\end{aligned}
\tag{5}
$$

Furthermore, for any belief function $b : H_t^o \to \mathcal{B}$ such that if $b(H_{t'}^o) = b(H_{t''}^o)$ then $p(s_t \mid H_{t'}^o) = p(s_t \mid H_{t''}^o)$ we have

$$
\max_{\pi(a|H_t^o)} V_{\pi}(H_t^o) \quad = \max_{\pi(a|b(H_t^o))} V_{\pi}(H_t^o), \forall H_t^o.
\tag{6}
$$

Moreover, $p(s_t|H_t^o)$ can be computed recursively, meaning that $\exists f : p(s_t|H_t^o) = f(p(s_{t-1}|H_{t-1}^o), a_{t-1}, o_t, r_t)$, which motivates the following alternative problem:

$$
\begin{aligned}
b_t &= f(b_{t-1}, a_{t-1}, o_t, r_t), \quad a_t \sim \pi^*(a|b_t), \\
\pi^*, f^* &= \arg\max_{\pi \circ f} V_{\pi}(b(H_t^0)), \quad \forall H_t^0.
\end{aligned}
\tag{7}
$$

Note that $b_t \in \mathcal{B}$ is computed from the observed history, that is $b_t = b(H_t^o)$. The solution to Problem 7 is a lower bound of Problem 3, and if $b(\cdot)$ satisfies the condition in Remark 1 then equality is reached. This condition is satisfied in the particular case were $b(H_t^o) = H_t^o$, implying that a belief that encodes the observed history solves Problem 3. If the conditional entropy $\mathbf{H}(s|H_t^o) \to 0 \; \forall H_t^o$, the state of the system is determined by the trajectory and Problem 3 is equivalent to Problem 2.

## 4 Instance learning and generalization in RL

In many RL scenarios the agent learns its policy by interacting with a finite number of levels [12]; to model and analyze this scenario, we propose a dual formulation of the POMDP dynamics as instances. Instances are repeatable, non-random, action-sequence-dependent samples of the transition model. We define the generation process of an instance, and later analyze the transition dynamics an agent observes when interacting with a finite set of instances. We then show how the original POMDP dynamics are related to the instance set dynamics, and describe how the value of a policy over an instance set relates to the value of the same policy over the POMDP.

**Definition 4.1. Environment Instance.** An instance $i$ is defined by a deterministic trajectory function that takes in a sequence of actions $a_{0:t-1} \in \mathcal{A}^{\otimes t}$ and outputs the sequence of visited states, observations, and rewards $\boldsymbol{\tau}_t^i : \mathcal{A}^{\otimes t} \to (\mathcal{S}^{\otimes t+1}, \mathcal{O}^{\otimes t+1}, \mathcal{R}^{\otimes t})$. This function is defined by the following (recursive) generation process,

$$s_0^i \sim \mu, k^i \sim U_{|\mathcal{K}|}, o_0^i \sim O(o|s_0^i, k^i),$$
$$\boldsymbol{\tau}_t^i(a_{0:t-1}) := \boldsymbol{\tau}_t^i(a_{0:t-2}) \oplus (s_t^i, o_t^i, r_t^i)|\boldsymbol{\tau}_{t-1}^i(a_{0:t-2}), a_{t-1}, \quad \forall a_0^{t-1} \in \mathcal{A}^{\otimes t}, \quad (8)$$
$$(r_t^i, s_t^i, o_t^i)|\boldsymbol{\tau}_{t-1}^i(a_{0:t-2}), a_{t-1} \sim T(r, s \mid a_{t-1}, s_{t-1}^i)O(o|a_{t-1}, s, k^i).$$

The initial states and observations $(s_0^i, o_0^i)$ of an instance $i$ are action independent. Variable $k^i$ captures the randomness of the observation distribution for an instance $i$ and is assumed to have a finite support ($k^i \in \mathcal{K} : |\mathcal{K}| < \infty$).

An instance provides consistent trajectories for any possible action sequence effected on (i.e., if the agent tries the same action sequence twice on the same instance, it will get the same result), this is shown graphically in Figure 1. We can map all possible trajectories in an instance to a tree structure where each node is completely determined by the action sequence. The trajectory functions $\boldsymbol{\tau}_t^i$ are equivalent to samples in the supervised learning setting, and independent of the learned policy.

An agent learns its policy from interacting with a finite set of instance transition functions $\boldsymbol{\tau}_t^i, i \in I$. On each episode, an instance $i$ is sampled uniformly from set $I$, the agent then interacts with this instance until the episode terminates (that is, instances are consistent along the episode and are sampled independently of the policy). The collected experience is then $(a_{0:T_n-1}, o_{0:T_n}^i, r_{1:T_n}^i)$ where $T_n$ is the length of the episode, $a_{0:T_n-1}$ is the action sequence, and $o_{0:T_n}^i, r_{1:T_n}^i$ are the corresponding observations and rewards according to $\boldsymbol{\tau}_t^i$. Note that the agent does not observe the state directly.

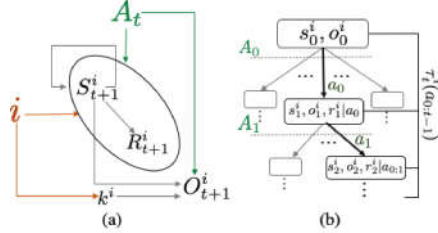

Figure 1: (a) Dependency graph for instance function $i$ following Definition 4.1. (b) Slice of instance trajectory tree $\boldsymbol{\tau}^i(\cdot)$ along action sequence $a_{0:t-1}$.

If the action set and the episode length are both finite ($|\mathcal{A}| \leq \infty, T_n \leq \bar{T}$), a single instance can produce up to $|\mathcal{A}|^{\bar{T}}$ distinct trajectories. We will abuse the notation $\boldsymbol{\tau}_t^i(a_{0:t-1})$ to indicate both the full output of the instance function or the current node in the trajectory tree (see Figure 1.b).

### 4.1 Markov property of instance sets

We study how the state and reward transition matrix of a set of instances $I$ differs from the one corresponding to the base environment POMDP ($T(r, s|a_t, s_t)$ in Equation 1). From Eq. 8 we observe that the future state and reward of an instance $i$ depends on the node of the transition tree $\boldsymbol{\tau}_t^i(a_{0:t-1})$ and action $a_t$. Given a history $H_t = \{a_{0:t-1}, s_{0:t}, o_{0:t}, r_{1:t}\}$ the stochasticity in the transition matrix of future state and reward for an instance set $I$ ($T^I(r, s|H_t, a_t)$) is only due to the uncertainty in determining to what node of what instance $H_t$ was collected from,[3] this can be written as

$$r_{t+1}, s_{t+1} \mid H_t, a_t, I \quad \sim T^I(r, s \mid H_t, a_t),$$
$$T^I(r, s \mid H_t, a_t) \quad = \mathbb{E}_{i \mid H_t, I}[T^i(r, s \mid \boldsymbol{\tau}_t^i(a_{0:t-1}), a_t)], \tag{9}$$
$$T^i(r, s \mid \boldsymbol{\tau}_t^i(a_{0:t-1}), a_t) \quad = \delta\big((r, s) = r_{t+1}^i, s_{t+1}^i \mid \boldsymbol{\tau}_{t+1}^i(a_{0:t})\big).$$

This shows that for a particular instance $i$, the state and reward transition matrix $(T^i(.\mid H_t, a_t))$ defines a deterministic distribution and only depends on the full sequence of actions, not on the current state of the system, as would be the case in the true game dynamics. Moreover, over a finite set of independent instances $I$, the Markov property is not satisfied by just the action sequence, but by the entire $H_t$. The latter is used to implicitly infer which instances $i \in I$ are compatible with the observed history $(p(i \mid H_t, I) > 0$, meaning that the trajectory function $i$ contains a branch that matches $H_t$).

Fortunately, if we take expectation on the sets $I$ of independently generated instances $i$ we recover the transition model of the environment, which indicates that a sufficiently large instance set represents the true model dynamics, as shown in Lemma 1. All proofs are found in Section A.1; a glossary is provided in Section A.2.

**Lemma 1. Expected instance transition matrix.** Given a set $I$ of $n$ independent instances $I \sim U_{\mathbb{N}}^{\otimes n}$, we have that $\forall H_t$ ($t < \infty$) compatible with $I$,

$$\mathbb{E}_{I \mid H_t}[T^I(r, s \mid H_t, a_t)] = \mathbb{E}_{I \mid H_t} \mathbb{E}_{i \mid H_t, I}[T^i(r, s \mid H_t, a_t)] = T(r, s \mid a_t, s_t), \quad \forall r, s. \tag{10}$$

We can conclude that the Markov property of the transition matrix given the full past trajectory goes from depending on the full action sequence (case of a particular instance $i$), to depending on the history (case of a finite set of independent instances $I$), and as the number of instances continues to grow, the transition matrix becomes indistinguishable from the original POMDP transition matrix that depends only on the previous state and action.

Although the transition matrix is not explicitly estimated in the model-free RL setting, it plays a central role in the process of optimizing the policy. Since we wish to generalize to unseen levels, we want to carefully ignore all policy dependencies that are not strictly state dependent, and ignore all extra information in $H_t$ that could be used to infer $i$ and $\boldsymbol{\tau}_t^i$. If we had access to the states $S$ directly, we could enforce the policies to only be state dependent. However, in a POMDP the states are unobserved, and we may need to capture temporal information in $H_t^o$ to infer $S$, leading to a conflict between inferring the state $S$ and not inferring the instance $i$ and its trajectory function. We formalize this in the following sections.

## 4.2 Value function and optimal policy

In a POMDP, an actionable policy can only depend on observed histories $H_t^o = \{a_{0:t-1}, o_{0:t}, r_{1:t}\}$. We define the value of a trajectory-dependent policy over a set of instances $V_\pi^I(H_t^o)$, and in Lemma 2 show that it is an unbiased estimator of the POMDP value function $V_\pi(H_t^o)$.

**Definition 4.2. Instance Value Function.** The value of an observable trajectory $H_t^o$ over a set of independently generated instances $I \sim U_{\mathbb{N}}^{\otimes n}$ according to policy $\pi : (\mathcal{A}^{\otimes t-1}, \mathcal{O}^{\otimes t}, \mathcal{R}^{\otimes t-1}) \to \mathcal{A}$ is

$$V_\pi^I(H_t^o) := \mathbb{E}_{R_{t+1:T} \mid H_t^o, \pi, I}[\textstyle\sum_{j=1}^{t-T} \gamma^{j-1} R_{t+j}] = \mathbb{E}_{\substack{A \sim \pi(\cdot \mid H_t^o) \\ R, O \sim p(.\mid H_t^o, A, I)}} [R + \gamma V_\pi^I(H_t^o \oplus (A, O, R))].$$

The reward and observation expectations at time $t + 1$ are taken over the distribution

$$p(r, o \mid H_t^o, A, I) = \textstyle\sum_{i, \boldsymbol{\tau}_t, k} \sum_{s_{t+1}} O^i(o \mid s_{t+1}, k, A, \boldsymbol{\tau}_t) T^i(s_{t+1}, r \mid \boldsymbol{\tau}_t, A) p(i, \boldsymbol{\tau}_t, k \mid H_t^o, I).$$

Here $p(i, \boldsymbol{\tau}_t, k \mid H_t^o, I)$ is the joint distribution of instance $i$, trajectory $\boldsymbol{\tau}_t$, and observation variable $k$ conditioned on instance set $I$ and observed trajectory $H_t^o$.

**Lemma 2. Unbiased value estimator.** Given a set $I$ of $n$ independent instances $I \sim U_{\mathbb{N}}^{\otimes n}$ and policy $\pi$, we have that $\forall H_t^o$ ($t < \infty$) compatible with $I$, $\mathbb{E}_{I \mid H_t^0}[V_\pi^I(H_t^o)] = V_\pi(H_t^o)$.

Although for a given policy $\pi$ $V_\pi^I(H_t^o)$ is an unbiased estimator of $V_\pi(H_t^o)$, the underlying transition matrices $T^I$ and $T$ have significant differences when $|I|$ is not large enough, this is reflected in the optimal belief function that, for a finite set $I$, will tend to overspecialize; once we are learning on a specific set of instances, the function that processes histories can ignore information that is relevant

for generalization, and expose information that is beneficial for improved performance on the set of instances. The following lemma shows that a belief that captures the state distribution (and thus generalizes to unseen instances) is potentially sub-optimal on the training instance set $I$.

**Lemma 3. State belief sub-optimality.** Given a finite set of instances $I \sim U_{\mathbb{N}}^{\otimes n}$ and a belief function such that $b^I(H_t^o) = p(i, \boldsymbol{\tau}_t | H_t^o, I), \forall H_t^o$:

$$\pi^I = \underset{\pi(a|b^I(H_t^o))}{\arg\max} \; V_\pi^I(b^I(H_t^o)) \in \underset{\pi(a|H_t^o)}{\arg\max} V_\pi^I(H_t^o). \tag{11}$$

Moreover, a policy that depends on the generalizeable belief function $b(H_t^o) = p(s_t | H_t^o)$ is potentially sub-optimal for $I$. Conversely, the policy $\pi^I$ is potentially sub-optimal for the true value function $V_\pi(H_t^o)$:

$$\begin{aligned} \underset{\pi(a|b^I(H_t^o))}{\max} V_\pi^I(b^I(H_t^o)) &\geq \underset{\pi(a|b(H_t^o))}{\max} V_\pi^I(b(H_t^o)), \\ \underset{\pi(a|b(H_t^o))}{\max} V_\pi(b(H_t^o)) &\geq V_{\pi^I}(b^I(H_t^o)). \end{aligned} \tag{12}$$

As mentioned before, Lemma 3 shows that the learned policy and belief functions are prone to suboptimality. A simple example where the differences between these two policies can be large is shown in Section A.3; the empirical evidence in our experiments suggest that this phenomena also occurs elsewhere. We can apply the generalization results provided by [26] to bound the expected generalization error of the value a policy trained on a set of instances $I$, this bound depends on the mutual information between the instance set and the learned policy. This shows that decreasing the dependence of a policy on the instance set tightens the generalization error.

**Lemma 4. Generalization bound on instance learning.** For any environment such that $|V_\pi(H_t^o)| \leq C/2, \forall H_t^o, \pi$, for any instance set $I$, belief function $b$, and policy function $\pi(b(H_o^t))$, we have

$$\underset{I}{\mathbb{E}} |V_\pi^I(\emptyset) - V_\pi(\emptyset)| \leq \sqrt{\frac{2C^2}{|I|} \times \mathrm{MI}(I, \pi \circ b)}, \tag{13}$$

with $\emptyset$ indicating the value of a recently initialized trajectory before making any observation.

The results in lemmas 3 and 4 suggest that the objective of maximizing policy value over the training set is misaligned with the true goal of learning a generalizeable policy; this generalizeable policy is sub-optimal in the sense that it can be improved with knowledge of the specific instance to which it is being applied to. In the following section we propose a method to tackle this problem. The experiments and result section provide empirical backing to the observations made so far.

## 5 Instance agnostic policy with ensembles

Our goal is to learn a policy that generalizes to multiple instances drawn from the same underlying dynamics. Ideally, this policy should capture the state distribution given the observed trajectory, $p(s_t | H_t^o)$. However, since states are unknown and training is done on a finite instance set, the belief representation may attempt to encode a distribution across instance-specific trajectories instead, as shown in Lemma 3. This phenomenon occurs because maximizing the value objective over the training set encourages speedrun-like policies.

We formulate a simple training scheme where we split our instances into subsets $\{I_m\}_{m=1}^M$. We assume these are large enough to contain a representative sample of possible hidden state transitions of the underlying model; this assumption is made so that each instance subset is potentially able to learn a state dependent policy that generalize well. In practice, we consider each distinct environment configuration and starting condition to be a distinct instance.

Following the Bayesian and information theoretic multiple task sampling formulation in [30], the agent has a

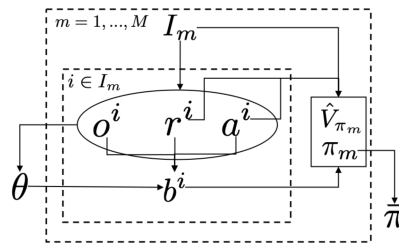

Figure 2: Simplified learning dependency scheme for the instance-agnostic policy ensembles approach. Variables $o^i, r^i, a^i$ represent collected observations, rewards and actions for an instance $i$ in subset $I_m$ (interdependecies are omitted). Shared parameter $\theta$ is learned from all instances, subset-specific policies and values $(\pi_m, \hat{V}_{\pi_m})$ are learned from rewards, actions, and shared representation $(b^i)$ within each instance $i \in I_m$. The agnostic policy $\bar{\pi}$ is the average over $\pi_m$.

shared representation $b_\theta \in \mathbb{R}^{|\mathcal{B}|}$, described by a learnable recursive function $f_\theta : \mathcal{O} \times \mathbb{R} \times \mathcal{A} \times \mathbb{R}^{|\mathcal{B}|} \to \mathbb{R}^{|\mathcal{B}|}$ parametrized by $\theta$ such that $b_{\theta,t} = f_\theta(o_t, r_t, a_{t-1}, b_{\theta,t-1})$. This shared representation is followed by an instance-subset-specific policy function $\pi_m : \mathbb{R}^{|\mathcal{B}|} \to \Delta^{|\mathcal{A}|}, \forall m$, which encourages part of the eventual instance-specific specialization to occur at the policy level, instead of on $f_\theta$. We also estimate the subset-specific value function $V_{\pi_m}$ with $\hat{V}_{\pi_m} : \mathbb{R}^{|\mathcal{B}|} \to \mathbb{R}, \forall m$, required for policy improvement.

We define the consensus policy $\bar{\pi}(a|b_\theta) := \sum_m \pi_m(a|b_\theta)/M$ as the average over the subset policies evaluated on the shared representation; this policy is used to collect training trajectories, and is also the policy used for any new, unseen level. The joint learning objective is

$$\max_{\theta,\{\pi_m\}} \frac{1}{M} \sum_{m=1}^M \frac{1}{|I_m|} \sum_{i \in I_m} \mathbb{E}_{H_t^o|i,\bar{\pi}} \left[ \hat{V}_{\pi_m}(b_{\theta,t}|H_t^o) \right] - \lambda ||\theta||_2^2,$$

$$\min_{\theta,\{\hat{V}_{\pi_m}\}} \mathbb{E}_{H_t^o|i,\bar{\pi}} \left[ ||\hat{V}_{\pi_m}(b_{\theta,t}|H_t^o) - V_{\pi_m}(H_t^o)||_2^2 \right], \tag{14}$$

where $V_{\pi_m}(H_t^o)$ denotes the true policy value, and is approximated through sampled returns (see Section A.4). A simplified illustration of how these variables are related is shown in Figure 2. Using the consensus policy for data collection may prevent, as confirmed in our experiments, the exploitation of speedrun-like trajectories during training (e.g., taking a running jump with the foreknowledge that no hazard will be present at the landing location), unless these actions generalize well across instance subsets. A shared representation is consistent with the POMDP formulation and encourages knowledge transfer between different observational distributions (parameter $k$ in Figure 1), leading to a representation that better captures the underlying state dynamics in its decision-making process. Adding an $\ell_2$ prior to representation parameter $\theta$, combined with the instance-set-specific policies $\pi_m$, disincentives the representation from encoding instance-specific information. Reducing the information the representation captures about the instance set also promotes generalization as seen in [26] and shown in Lemma 4. To make use of samples taken from the consensus policy $\bar{\pi}$ on instance specific policies and values, we use importance weights and perform off-policy policy gradients; implementation details are provided in Section A.4.

# 6 Experiments and results

The goal of the experiments presented next is to support the theoretical foundations presented in Section 3. Such theory explains well known empirical observations and motivates the algorithm in Section 5. These experiments stress how conscientious modelling of our usual training processes opens the door to a better understanding of RL and the potential development of new computational tools. We use the popular CoinRun environment [12], a 2D scrolling platformer with 7 distinct actions, where the agent observes a $64 \times 64$ RGB image and the only source of non-zero reward is when it reaches a golden coin at the end of the level, with a reward of 10. Following [12], we analyze the generalization benefits of a baseline method only using BatchNorm [40] (base), additional $\ell_2$ regularization ($\ell_2$), and Cutout [37] ($\ell_2$-CO). We compare these with the performance of our proposed *instance agnostic policy ensemble* (IAPE) method, and a model trained on an unbounded level set ($\infty$-levels), considered as the gold standard in terms of generalization. We use the Impala-CNN architecture [41] followed by a single 256-unit LSTM, policy and value functions are implemented as single dense layers. Agents are trained on a set of 500 levels for a total of $256M$ frames; testing is done on an unbounded level set different from the one used to train the $\infty$-levels agent. Details on implementation, evaluation, and extended results are provided in Section A.5.

## 6.1 Generalization performance

We evaluate the performance of the described methods and the proposed IAPE by measuring the episode return ($R$), the difference between time-to-reward on successful instances per level w.r.t the base model ($\Delta T_{base}|R = 10$), as well as the per-instance KL-divergence between the time-averaged policy of each method and the one obtained on the unbounded training levels ($D_{kl}^i(\pi_\infty|\pi)$). The $\Delta T_{base}|R = 10$ is an indicator of how speed-run-like policies are when compared to the baseline policy, with higher values indicating more cautious agents. Lower values in the $D_{kl}^i(\pi_\infty|\pi)$ metric may indicate that the learned policy is closer to the unbounded level policy.

Table 1 summarizes these results, as expected the baseline model has the highest train return and lowest test return. Moreover, the empirical distribution of $\Delta T_{base}|R = 10$ is positive for every method, which is consistent with the baseline model learning speed-running policies. We show these

empirical distributions in Supplementary Material, in most cases these are positively skewed, this is most evident on training levels and for the $\infty$-levels policy. The baseline model differs considerably in terms of $D_{kl}^i(\pi_\infty|\pi)$ in both training and validation, indicating that this policy differs significantly from the ideal ($\infty$-levels) model, this follows from our theoretical analysis.

Adding different types of regularization generally prevents overfitting and improves rewards on unseen levels. This also reduces the average divergence between these policies and the unbounded level policy ($D_{kl}^i(\pi_\infty|\pi)$), indicating that the regularized policies are closer to the desired one. The proposed IAPE model has the best performance both in terms of rewards and distance to the optimal policy $D_{kl}^i(\pi_\infty|\pi)$. Additionally, Figure 3.a shows how each model adapts to new training levels, and how does their performance on the previous training and test set is affected. IAPE is consistently better in maintaining performance on old levels and specializing to the new dataset.

Table 1: Performance comparison on CoinRun benchmark, average episode length varies between 45 to 50 frames. IAPE is the best performing method on test levels; $\infty$-levels is the target policy.

| Method | base | $\ell^2$ | $\ell^2$-CO | IAPE | $\infty$-levels |
|---|---|---|---|---|---|
| $R$ train | **9.74$\pm$.09** | 9.56$\pm$.14 | 9.36$\pm$.15 | 9.54$\pm$.13 | 9.14$\pm$.14 |
| $D_{kl}^i(\pi_\infty|\pi)$ train | 0.57$\pm$0.39 | 0.2$\pm$0.16 | 0.15$\pm$0.12 | **0.11$\pm$0.12** | - |
| $\Delta T_{base}|$R=10 train | - | 1.34$\pm$7.94 | 3.3$\pm$8.79 | 2.73$\pm$8.15 | 4.65$\pm$8.39 |
| $R$ test | 7.47$\pm$.25 | 7.49$\pm$.22 | 7.99$\pm$.20 | **8.23$\pm$.18** | **9.05$\pm$.23** |
| $D_{kl}^i(\pi_\infty|\pi)$ test | 0.57$\pm$0.39 | 0.19$\pm$0.15 | 0.15$\pm$0.11 | **0.1$\pm$0.09** | - |
| $\Delta T_{base}|$R=10 test | - | 1.73$\pm$12.71 | 0.45$\pm$11.54 | 1.14$\pm$12.28 | 1.51$\pm$12.49 |

## 6.2 Ensemble analysis

We compare how the exploration strategy in IAPE affects its learned parameters and generalization capabilities by comparing against a similar ensemble baseline (EB) paradigm were the experience for each instance subset is collected with its specific policy (and not the agnostic policy). We measure the difference in the instance-subset-specific parameters with pairwise cosine similarity between weight matrices of different ensembles for both policy and value layer parameters.

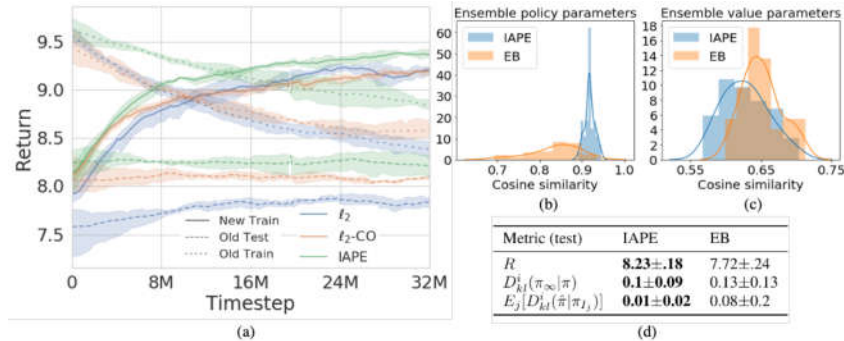

Figure 3: a) Continual learning performance of $\ell_2$, $\ell_2$-CO and IAPE on a new set of 500 levels, trained for $64M$ timesteps, performance on old training and test levels is also shown. b) & c) Cosine similarity between policy and value parameters respectively for IAPE and EB. d) Performance comparison.

Figures 3.b and 3.c show the cosine similarity distribution between policy and value parameters across ensembles for the proposed IAPE and EB. Similarity between policy parameters is well concentrated around 0.92 for the IAPE method, something that is not true on the EB approach. Since the only difference between these two is how the experience is collected during training, it is reasonable to conjecture that this data collection procedure regularizes the subset-specific policies to be similar to the agnostic policy. Value-specific parameters in Figure 3.c show higher diversity across both methods; this is expected since values may differ significantly between instances, regardless of policy. Table 3.d shows that IAPE has better test-time performance than EB, being closer to the unbounded level policy. It also shows greater agreement between instance-specific policies and its own policy

average, measured as the average KL divergence between each instance subset and its corresponding agnostic policy ($E_j[D_{kl}^i(\hat{\pi}|\pi_{I_j})]$), this result is consistent with the one shown in Figure 3.b.

## 7 Discussion

We introduced and formalized instances in the RL setting; instances are deterministic mappings from actions to states, observations, and rewards consistent with the standard POMDP formulation. We showed that instances directly impact the effective state space of the environment the agent learns from, which may induce any agent to fail to capture the true, generalizable dynamics of the environment. This problem is exacerbated in the POMDP scenario, where we use past observations to infer the current state, but those can also induce overfitting to the training instances. We show that the optimal policy for a finite instance set is a speed-run one that learns to identify the instance, memorizing the optimal sequence of actions for its specific trajectory mapping. This policy differs from the desired one, which should only capture the underlying dynamics of the environment.

We empirically compare regularization methods with the proposed IAPE, and show that results align with the presented theory. In particular, the policy of an agent trained naively on a finite number of levels considerably differs from one that is trained on an unbounded level set; this gap is reduced with the use of regularization. The proposed instance agnostic policy ensemble showed promising results on unseen levels. Moreover, we showed that collecting experience from the agnostic policy during training had a large positive impact in the performance of the method.

There are significant theoretical and practical benefits to be obtained from explicit consideration of the difference between training set dynamics and environment dynamics. Future work should focus on more targeted methods to account for these differences in dynamics via regularization.

## Broader Impact

Understanding generalization properties in reinforcement learning (RL) is one of the most critical open questions in modern machine learning, with implications ranging from basic science to socially-impactful applications. Towards this goal, we formally analyze the training dynamics of RL agents when environments are reused. We prove that this standard RL training methodology introduces undesired changes in the environment dynamics, something to be aware of since it directly impacts the learned policies and generalization capabilities. We then introduce a simple computational methodology to address this problem, and provide experimental validation of the theory presented. Beyond the scope of this paper, deep reinforcement learning has multiple human-facing applications, but is notoriously data inefficient and more generalization understanding is needed. Addressing these fundamental problems and providing a foundational understanding is critical to increase its real-world applicability in fields such as robotics, chemistry, and healthcare. This work is a step in this direction, with building blocks that will encourage and facilitate future critical developments.

## Acknowledgements

Work partially supported by NSF, NGA, ARO, ONR, and gifts from Cisco, Google, Amazon, Microsoft, and Intel AI Lab.

## Footnotes

[2]For brevity we denote $T(r, s \mid a_t, s_t) = p(r \mid a_t, s) p(s \mid a_t, s_t)$.

[3]Note that on the original POMDP we have $T(r, s|a_t, H_t) = T(r, s|a_t, s_t)$.

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
