[Supplementary Material]

# A   Supplementary Material

## A.1   Proofs

**Lemma 1. Expected instance transition matrix.** Given a set $I$ of $n$ independent instances $I \sim U_{\mathbb{N}}^{\otimes n}$, we have that $\forall H_t$ ($t < \infty$) compatible with $I$,

$$\mathbb{E}_{I|H_t}[T^I(r, s|H_t, a_t)] = \mathbb{E}_{I|H_t} \mathbb{E}_{i|H_t, I}[T^i(r, s|H_t, a_t)] = T(r, s \mid a_t, s_t), \quad \forall r, s. \quad (15)$$

*Proof.* We observe that

$$
\begin{aligned}
\mathbb{E}_{I|H_t}[T^I(r, s|H_t, a_t)] &= \mathbb{E}_{I|H_t}[\mathbb{E}_{i|H_t, I}[T^i(r, s|H_t, a_t)]], \\
&= \mathbb{E}_{I|H_t}[\mathbb{E}_{i|H_t, I}[\delta((r^i, s^i) = (r, s)|H_t, a_t, i)]], \\
&= \mathbb{E}_{i|H_t}[\delta((r^i, s^i) = (r, s)|H_t, a_t, i)], \\
&= \mathbb{E}_{(r^i, s^i)|H_t}[\delta((r^i, s^i) = (r, s)|H_t, a_t)], \\
&= T(r, s \mid a_t, s_t),
\end{aligned}
\quad (16)
$$

where the first equality is taken from Equation 9 and we leveraged the construction process of the transition matrix for the instance $T^i$ and the basic property $\mathbb{E}[\delta(x = x_0)] = P(x = x_0)$.

$\square$

The following corollary (Corollary 1) states that the expectation over the instances of the *instances-specific* probability distribution of future rewards, states and observations for every past history $H_t$ and policy $\pi$ $(p((r, s, o)_{t+1:t+n}|H_t, \pi, I))$, is the transition distribution of the future rewards, states and observations of the environment $T((r, s, o)_{t+1:t+n}|H_t, \pi)$. This result will be used to prove Lemma 2.

**Corollary 1.** Given a set $I$ of $n$ independent instances $I \sim U_{\mathbb{N}}^{\otimes n}$, a policy $\pi(a \mid H^t)$, and a time horizon $n$, we have that $\forall H_t$ ($t < \infty$) compatible with $I$

$$
\begin{aligned}
&\mathbb{E}_{I|H_t}[p((r, s, o)_{t+1:t+n}|H_t, \pi, I)] = T((r, s, o)_{t+1:t+n}|H_t, \pi), \forall r, s, \\
&T((r, s, o)_{t+1:t+n}|H_t, \pi) := \int_{a_{t:t+n-1}} \Pi_{j=0}^{n-1} \pi(a_{t+j} \mid H_{t+j}) T((r, s, o)_{t+j+1} \mid s_{t+j}, a_{t+j}).
\end{aligned}
\quad (17)
$$

Where (marginalizing over observational variable $k$ in Equation 1 for simplicity) we have

$$T((r, s, o)_{t+j+1} \mid s_{t+j}, a_{t+j}) := T((r, s)_{t+j+1} \mid s_{t+j}, a_{t+j}) O(o_{t+j+1} \mid a_{t+j}, s_{t+j+1}). \quad (18)$$

*Proof.* Following the proof of Lemma 1, we observe that for any sequence of rewards, states and observations $\bar{r} = r_{t+1:t+j}, \bar{s} = s_{t+1:t+j}, \bar{o} = o_{t+1:t+j}$, and any sequence of actions $\bar{a} = a_{t:t+j-1}$, the following holds

$$
\begin{aligned}
\mathbb{E}_{I|H_t}[p(\bar{r}, \bar{s}, \bar{o} \mid H_t, \bar{a}, I)] &= \mathbb{E}_{I|H_t}[\mathbb{E}_{i|H_t, I}[p(\bar{r}, \bar{s}, \bar{o} \mid H_t, \bar{a}, i)]], \\
&= \mathbb{E}_{I|H_t}[\mathbb{E}_{i|H_t, I}[\delta((\bar{r}^i, \bar{s}^i, \bar{o}^i) = (\bar{r}, \bar{s}, \bar{o})|H_t, \bar{a}, i)]], \\
&= \mathbb{E}_{\bar{r}^i, \bar{s}^i, \bar{o}^i}[\delta((\bar{r}^i, \bar{s}^i, \bar{o}^i) = (\bar{r}, \bar{s}, \bar{o})|H_t, \bar{a}, i)], \\
&= T(\bar{r}, \bar{s}, \bar{o} \mid \bar{a}, s_t) = \Pi_{j=0}^{n-1} T((r, s, o)_{t+j+1} \mid s_{t+j}, a_{t+j}).
\end{aligned}
\quad (19)
$$

That is, the expectation across instances of the $n$-step transition of states, rewards, and observations of the instance transition model matches the underlying model for any sequence of actions.

On the other hand, if we consider a particular policy $\pi$ we have

$$
\begin{aligned}
\mathbb{E}_{I|H_t}[p(\bar{r}, \bar{s}, \bar{o} \mid H_t, \pi, I)] &= \mathbb{E}_{I|H_t}[\int_{\bar{a}} \Pi_{n=0}^{j-1} \pi(a_{t+n} \mid H_{t+n}) p(\bar{r}, \bar{s}, \bar{o} \mid H_t, \bar{a}, I)], \\
&= \int_{\bar{a}} \mathbb{E}_{I|H_t}[\Pi_{n=0}^{j-1} \pi(a_{t+n} \mid H_{t+n}) p(\bar{r}, \bar{s}, \bar{o} \mid H_t, \bar{a}, I)], \\
&= \int_{\bar{a}} \Pi_{n=0}^{j-1} \pi(a_{t+n} \mid H_{t+n}) \mathbb{E}_{I|H_t}[p(\bar{r}, \bar{s}, \bar{o} \mid H_t, \bar{a}, I)], \\
&= \int_{\bar{a}} \Pi_{n=0}^{j-1} \pi(a_{t+n} \mid H_{t+n}) T(\bar{r}, \bar{s}, \bar{o} \mid \bar{a}, s_t), \\
&= T(\bar{r}, \bar{s}, \bar{o} \mid H_t, \pi).
\end{aligned}
\quad (20)
$$

On the first equality we marginalize across actions, on the second equality we exchange the integral and the expectation operator, and on the third we observe that $\Pi_{n=0}^{j-1} \pi(a_{t+n} \mid H_{t+n})$ is constant w.r.t. $I$. The fourth and final equality are derived from equations 19 and 15 respectively.

This shows that the expected (across instance sets) reward, state, and observation transition matrix for any policy $\pi$ and history $H^t$ matches the true model. $\qquad\square$

**Lemma 2. Unbiased value estimator.** Given a set $I$ of $n$ independent instances $I \sim U_{\mathbb{N}}^{\otimes n}$, and policy $\pi$, we have that $\forall H_t^o$ $(t < \infty)$ compatible with $I$,

$$\mathbb{E}_{I|H_t^o}[V_\pi^I(H_t^o)] = V_\pi(H_t^o). \tag{21}$$

*Proof.* We observe that

$$\mathbb{E}_{I|H_t^o}[V_\pi^I(H_t^o)] = \mathop{\mathbb{E}}_{I|H_t^o}[\mathop{\mathbb{E}}_{\substack{A \sim \pi \\ \{R_{t+j}\}_{j=1}^{T-t}|I,H_t^o,\pi}}[\textstyle\sum_{j=1}^{T-t} \gamma^{j-1} R_{t+j}]]. \tag{22}$$

By linearity of expectation, we focus on $R_{t+j}$,

$$
\begin{aligned}
E_{I|H_t^o,\pi}[R_{t+j}] &= \int_{R_{t+j}} R_{t+j} \sum_I p(I \mid H_t^o) p(R_{t+j} \mid I, \pi, H_t^o), \\
&= \int_{S_{0:t}} P(S_{0:t} \mid H_t^o) \int_{R_{t+j}} R_{t+j} \sum_I p(I \mid H_t) p(R_{t+j} \mid H_t, \pi, I), \\
&= \int_{S_{0:t}} p(S_{0:t} \mid H_t^o) \int_{R_{t+j}} R_{t+j} \int_{\substack{R_{t+1}^{t+j-1} \\ (S,O)_{t+1}^{t+j}}} \sum_I p(I \mid H_t) p((R,O,S)_{t+1:t+j} \mid H_t, \pi, I), \\
&= \int_{S_{0:t}} p(S_{0:t} \mid H_t^o) \int_{(R,S,O)_{t+1}^{t+j}} R_{t+j} T((R,S,O)_{t+1:t+j} | H_t, \pi), \\
&= \int_{S_t} p(S_t \mid H_t^o) \int_{(R,S,O)_{t+1}^{t+j}} R_{t+j} T((R,S,O)_{t+1:t+j} | H_t^o, S_t, \pi), \\
&= \int_{R_{t+j}} R_{t+j} p(R_{t+j} | H_t^o, \pi) = E_{R_{t+j}|H_t^o,\pi}[R_{t+j}].
\end{aligned}
\tag{23}
$$

Here the second equality comes from marginalizing over $S_{0:t}$ and that $H_t = H_t^o \oplus S_{0:t}$, then we used Corollary 1 for the fourth equality. In the fifth equality we observe that the policy only depends on the observed history, and that the future trajectories depend on current state and the policy. From this result and the linearity of expectation, we recover the statement of the lemma.

$\qquad\square$

**Lemma 3. State belief sub-optimality.** Given a finite set of instances $I \sim U_{\mathbb{N}}^{\otimes n}$ and a belief function such that $b^I(H_t^o) = p(i, \tau_t | H_t^o, I), \forall H_t^o$,

$$\pi^I = \operatorname*{arg\,max}_{\pi(a|b^I(H_t^o))} V_\pi^I(b^I(H_t^o)) \in \operatorname*{arg\,max}_{\pi(a|H_t^o)} V_\pi^I(H_t^o). \tag{24}$$

Moreover, a policy that depends on the generalizeable belief function $b(H_t^o) = p(s_t | H_t^o)$ is potentially sub-optimal for $I$. Conversely, the policy $\pi^I$ is potentially sub-optimal for the true value function $V_\pi(H_t^o)$:

$$
\begin{aligned}
\max_{\pi(a|b^I(H_t^o))} V_\pi^I(b^I(H_t^o)) &\geq \max_{\pi(a|b(H_t^o))} V_\pi^I(b(H_t^o)), \\
\max_{\pi(a|b(H_t^o))} V_\pi(b(H_t^o)) &\geq V_{\pi^I}(b^I(H_t^o)).
\end{aligned}
\tag{25}
$$

*Proof.* Equation 24 is a straightforward application of Lemma 1, since from Equation 9 we observe that $i, \tau_t$ define the Markov kernel over $T^I$.

Equation 25 also follows from Lemma 1 since

$$
\begin{aligned}
\max_{\pi(a|b^I(H_t^o))} V_\pi^I(b^I(H_t^o)) &= \max_{\pi(a|H_t^o)} V_\pi^I(H_t^o), \\
&\geq \max_{\pi(a|b(H_t^o))} V_\pi^I(b(H_t^o)), \\
\max_{\pi(a|b(H_t^o))} V_\pi(b(H_t^o)) &= \max_{\pi(a|H_t^o)} V_\pi(H_t^o), \\
&\geq V_{\pi^I}(b^I(H_t^o)),
\end{aligned}
\tag{26}
$$

where the equalities are obtained from Lemma 1 and the inequalities from set inclusion, $\{\pi : H_t^o \to \Delta^{\mathcal{A}}\} \supseteq \{\pi : f(H_t^o) \to \Delta^{\mathcal{A}}\}$.

$\square$

**Lemma 4. Generalization bound on instance learning**. For any environment such that $|V_\pi(H_t^o)| \leq C/2, \forall H_t^o, \pi$, for any instance set $I$, belief function $b$, and policy function $\pi(b(H_o^t))$, we have

$$
\mathbb{E}_I |V_\pi^I(\emptyset) - V_\pi(\emptyset)| \leq \sqrt{\frac{2C^2}{|I|} \times \mathrm{MI}(I, \pi \circ b)},
\tag{27}
$$

with $\emptyset$ indicating the value of a recently initialized trajectory before making any observation.

*Proof.* Instances $I$ are independently sampled, since the value function $V_\pi^I$ of the model is bounded between $[-\frac{C}{2}, \frac{C}{2}]$ for any policy $\pi$, then $V_\pi^I$ is $C$-subGaussian $\forall I, \pi$. The result follows from observing Lemma 2 ($\mathbb{E}_I[V_\pi^I(\cdot)] = V_\pi(\cdot)$) and a direct application of Theorem 1 in [26]. $\square$

## A.2 Glossary

<div align="center">Table 2: Glossary table</div>

| Symbol | Name | Notes |
|---|---|---|
| $s \in \mathcal{S}$ | Environment state | |
| $a \in \mathcal{A}$ | Agent action | |
| $r \in \mathbf{R}$ | Instantaneous reward | |
| $o \in \mathcal{O}$ | Environment observation | |
| $k \in \mathcal{K}$ | Observation modality | Parameter affecting observations consistently throughout the episode, but independent of state transitions (e.g.: background color, illumination conditions) |
| $\mu$ | Initial state distribution | |
| $O(o \mid s, k)$ | Observation distribution | Observations in the POMDP are sampled from this distribution at each timestep |
| $T(r, s \mid a_t, s_t)$ | POMDP transition matrix | $T(r, s \mid a_t, s_t) = p(r \mid a_t, s)p(s \mid a_t, s_t)$, reward only depends on current state and action |
| $H_t = \{a_{0:t-1}, s_{0:t}, o_{0:t}, r_{1:t}\}$ | History | Collection of all relevant variables throughout an episode |
| $H_t^o = \{a_{0:t-1}, o_{0:t}, r_{1:t}\}$ | Observable history | Collection of all variables obseved by the agent |
| $\pi(a \mid s)$ | Agent policy | Depending on context, policy may depend on POMDP states $s$, history $H_t$, or observable history $H_t^0$ |
| $\Delta^{\mathcal{A}}$ | Simplex over $\mathcal{A}$ actions | Set of all possible distributions over $\mathcal{A}$ |
| $V_\pi(\cdot)$ | Value function | Value of policy $\pi$ conditioned on known factors $\cdot$ |
| $b : H_t^o \to \mathcal{B}$ | Belief function | Function that processes observed histories |
| $\mathbb{H}(\cdot)$ | Entropy | |
| $\tau_t^i : \mathcal{A}^{\otimes t} \to \left(\mathcal{S}^{\otimes t+1}, \mathcal{O}^{\otimes t+1}, \mathcal{R}^{\otimes t}\right)$ | Instance trajectory function | Deterministic function that characterizes an instance, assigns a history $H_t$ to any action sequence $\mathcal{A}^{\otimes t}$ (up to episode termination). Since $\tau_t^i(a_{0:t-1})$ uniquely defines a node in the instance trajectory tree, we abuse notation to also indicate current node on the trajectory tree |
| $T_n, \bar{T}$ | Episode lengths | Duration of $n$-th episode and maximum episode length respectively |
| $\delta(a = b)$ | Deterministic distribution | Assigns probability 1 to event $a = b$, and 0 everywhere else. |
| $T^i(r, s \mid a_t, H_t)$ | transition matrix of instance $i$ | instance transition matrix depends on entire history $H_t$, deterministic, $T^i(r, s \mid H_t, a_t) = \delta\left((r, s) = r_{t+1}^i, s_{t+1}^i \mid \tau_{t+1}^i(a_{0:t})\right)$. |
| $T^I(r, s \mid a_t, H_t)$ | transition matrix of instance set $I$ | instance set transition matrix depends on entire history $H_t$, stochasticity of the transition is a function of not knowing on which instance $i \in I$ the agent is acting on. |
| $V_\pi^I(H_t^o)$ | value function over instance set $I$ | value of policy $\pi$ conditioned on observed history $H_t^o$ and known instance set $I$ |
| $MI(\cdot, \cdot)$ | Mutual information | |

## A.3 Example of state belief sub-optimality

This example illustrates how optimal policies on true states and instances can have arbitrarily large differences both in behaviours and in expected returns. We present a sequential bandit environment, with a clear optimal policy, and we show that if we sample an instance from this environment, the optimal instance-specific policy for the environment is now trajectory-dependent, instead of state-dependent. The optimal environment policy is shown to have a significantly smaller return on a typical environment instance than its instance-specific counterpart. Conversely, the instance-specific policy has lower expected return on the true, generalizeable dynamics. We note that this is an intrinsic problem to instance learning (and reusing instances in general), and that this toy example showcases the statements made in Lemma 3.

Suppose we have a fully observable environment with a single state (bandit), and $|\mathcal{A}|$ actions, with the following state and reward transition matrix and observation function:

$$
\begin{aligned}
T(r=1, s \mid s, a=0) &= \overline{p}, \\
T(r=0, s \mid s, a=0) &= 1 - \overline{p}, \\
T(r=1, s \mid s, a \neq 0) &= \underline{p}, \\
T(r=0, s \mid s, a \neq 0) &= 1 - \underline{p}, \\
O(o \mid s, a) &= \delta(o = s).
\end{aligned}
\tag{28}
$$

Here $\overline{p} > \underline{p}$; an episode consists of $N$ consecutive plays. It is straightforward to observe that $p(s \mid H_t^o) = \delta_s, \forall H_t^o$, and therefore the state-distribution dependent policy $\pi(a \mid p(s \mid H_t^0))$ is constant $\pi(a \mid p(s \mid H_t^0)) = \pi_a \forall a, H_t^0$. Furthermore, the value of the initial observed history $H_t^0 = \emptyset$ can be computed as

$$
\begin{aligned}
V_{\pi(a \mid p(s \mid H_t^0))}(\emptyset) &= \mathbb{E}_{\pi(a \mid p(s \mid H_t^0))}[\textstyle\sum_{j=1}^N \gamma^{j-1} R_j], \\
&= (\pi_0 \overline{p} + (1 - \pi_0)\underline{p}) \textstyle\sum_{j=1}^N \gamma^{j-1}, \\
&= (\pi_0 \overline{p} + (1 - \pi_0)\underline{p}) \frac{1 - \gamma^N}{1 - \gamma},
\end{aligned}
\tag{29}
$$

and the maximal state-dependent policy is $\pi(a \mid p(s \mid H_t^0)) = \delta(a = 0)$ with value $V_{\delta(a=0)}(\emptyset) = \overline{p}\frac{1-\gamma^N}{1-\gamma}$.

On the other hand, suppose we have a single instance ($|I| = 1$) of this environment, with probability $(1 - \overline{p})(1 - (1 - \underline{p})^{\mathcal{A}-1})$ each node in the instance transition tree has zero reward on the optimal arm ($a = 0$), but non-zero reward on at least one sub-optimal arm. Overall, with probability $1 - ((1 - \overline{p})(1 - \underline{p})^{\mathcal{A}-1})$ each node in the instance tree has a non-zero reward action. It is thus straightforward to observe that a typical instance $i$ has an observation-dependent policy $\pi(a \mid H_t^0)$ that achieves

$$
\max_{\pi(a \mid H_t^0)} V_{\pi(a \mid H_t^0)}^I(\emptyset) \geq (1 - ((1 - \overline{p})(1 - \underline{p})^{\mathcal{A}-1})) \frac{1 - \gamma^N}{1 - \gamma}.
\tag{30}
$$

This return can be achieved by merely checking if the node in the instance tree the agent is on has any non-zero-reward action and selecting one of those at random. An illustration of the state dynamics of the environment versus the transition tree of the instance is shown on Figure 4.

Notice that as stated in Lemma 3, we have that the optimal *generalizeable* policy is sub-optimal on the instance set $I$, and vice-versa. Furthermore, for large action spaces $|\mathcal{A}|$, the instance-specific policy $\overline{\pi} = \arg\max_{\pi(a \mid H_t^0)} V_{\pi(a \mid H_t^0)}^I$ has an expected value on the instance set of $V_{\overline{\pi}}^I(\emptyset) = \frac{1 - \gamma^N}{1 - \gamma}$, but can be arbitrarily close to the worst possible return on the true environment $V_{\overline{\pi}}(\emptyset) \simeq \underline{p}\frac{1-\gamma^N}{1-\gamma}$. This undesired behaviour arises from a mismatched objective, where we want our policy to maximize expected reward on the model dynamics, but we instead provide instance-specific dynamics that might have different optimal policies.

Figure 4: Left: Transition model of a bandit with two actions. Right: transition tree of a given instance of the environment, the optimal action sequence is shown in red. Note that the optimal *instance-specific* policy takes different actions for observed histories with the same $p(s \mid H_t^0)$.

### A.4   Implementation details

The implementation details of the proposed instance agnostic policy ensembles method (IAPE) whose objective was described in Equation 14 are presented next. We describe the importance weighting technique used in order to leverage the experience acquired with the consensus policy to compute instance-specific policies and values. We then provide details about the architectures and hyperparameters.

Here we focus on Off-policy Actor-Critic (AC) techniques because we wish to make use of trajectories collected under one policy to improve another, we do this via modified importance weighting (IW). We use policy gradients for policy improvements, and similarly to [41, 42, 43], we modify the $n$-step bootstrapped value estimate for the off-policy case to estimate policy values.

Consider an observed trajectory $H_t^o$ collected using the consensus policy $\bar{\pi}$ on instance $i$ belonging to instance subset $I_m$. We use clipped IW to define the value target for policy $\pi_m$ at time $\tau$ as

$$
\begin{aligned}
g_\tau^m &:= \sum_{t=\tau}^{\tau+n-1} \gamma^{t-\tau} w_{m,\tau}^t r_{t+1} + \gamma^n w_{m,\tau}^{t+n} \hat{V}_{\pi_m}(b_{\theta,\tau+n}|H_{\tau+n}^o), \\
w_{m,\tau}^t &:= \text{clip}(\prod_{j=\tau}^t \tfrac{\pi_m(a_j|b_j)}{\bar{\pi}(a_j|b_j)}, \underline{w}, \bar{w}),
\end{aligned}
\tag{31}
$$

where $\underline{w} \le 1 \le \bar{w}$ define the minimum and maximum importance weights for the partial trajectory; this clipping is used as a variance reduction technique. The dependencies $g_\tau^m = g_\tau^m(H_t^o)$, $b_\tau = b(H_\tau^0)$, are omitted for brevity. Note that we clip the cumulative importance weight of the trajectory, since it is well reported that clipping $\frac{\pi_m(a_j|b_j)}{\bar{\pi}(a_j|b_j)}$ individually leads to high variance estimates (see [43]).

This clipping technique leads to the exact IW estimate for likely trajectories, and is equivalent to the on-policy $n$-step bootstrap estimate when both policies are identical. Using this estimator, the value target (critic) and policy (actor) losses for this sample are

$$
\begin{aligned}
l_{V_m}(H_t^0, \tau) &= ||\hat{\mathbf{V}}_{\pi_\mathbf{m}}(\mathbf{b}_{\theta,\tau}|\mathbf{H}_\tau^\mathbf{o}) - g_\tau||_2^2, \\
l_{\pi_m}(H_t^0, \tau) &= -\log(\pi_\mathbf{m}(\mathbf{a_j} \mid \mathbf{b_j})) \tfrac{\pi_m(a_j|b_j)}{\bar{\pi}(a_j|b_j)}(r_{\tau+1} + \gamma g_{\tau+1}^m - V_{\pi_m \circ b}(b_t)),
\end{aligned}
\tag{32}
$$

where the policy gradient also requires an importance weight similar to [41]. The bolded terms are the only gradient propagating terms in the loss. The full training loss for the model is computed as

$$
L = \mathbb{E}_{I_m|I} \mathbb{E}_{H_t^0,\tau|I_m,\bar{\pi}}[\tfrac{1}{2}l_{V_m}(H_t^0,\tau) + l_{\pi_m}(H_t^0,\tau)] + \lambda_{\text{reg}}||\theta, \{\phi_m\}, \{\psi_m\}, ||_2^2,
\tag{33}
$$

where $\lambda_{\text{reg}}$ is a prior over the network weights. Note that all instance-specific parameters only receive gradient updates from their own instance set.

All experiments use the Impala-CNN architecture [41] for feature extraction, these features are concatenated with a one-hot encoding of the previous action, and fed into a 256-unit LSTM, policy and value functions are implemented as single dense layers. Parameters for all experiments are shown in Table 3.

Table 3: Hyperparameter table.

| Method | base | $\ell^2$ | $\ell^2$-CO | IAPE | $\infty$-levels |
|---|---|---|---|---|---|
| $\gamma$ | .99 | .99 | .99 | .99 | .99 |
| bootstrap rollout length | 256 | 256 | 256 | 256 | 256 |
| $\underline{w}$ | 2 | 2 | 2 | 2 | 2 |
| $\overline{w}$ | $\frac{1}{2}$ | $\frac{1}{2}$ | $\frac{1}{2}$ | $\frac{1}{2}$ | $\frac{1}{2}$ |
| minibatch size | 8 | 8 | 8 | 8 | 8 |
| ADAM learning rate | $2 \times 10^{-4}$ | $2 \times 10^{-4}$ | $2 \times 10^{-4}$ | $2 \times 10^{-4}$ | $2 \times 10^{-4}$ |
| $\ell_2$ penalty | 0 | $2 \times 10^{-5}$ | $2 \times 10^{-5}$ | $2 \times 10^{-5}$ | $2 \times 10^{-5}$ |
| # of training levels | 500 | 500 | 500 | 500 | $\infty$ |
| uses Cutout | No | No | Yes | No | No |
| uses Batchnorm | Yes | Yes | Yes | Yes | Yes |
| # of ensembles | - | - | - | 10 | - |

## A.5 Extended results

Figures 5 and 6 show an extended version of the results presented in Table 1. Figure 5.a shows the empirical distribution of the time-to-reward difference on successful training instances w.r.t the base policy ($\Delta T_{base}|R = 10$) for each method. In most cases these distributions are positively skewed, indicating that the obtained policies tend to be slower than the baseline on training levels. This is considerably noticeable for the $\infty$-level policy. Figure 5.b presents the distribution of the per-training-instance KL-divergence between the time-averaged policy of each method and the one obtained on the unbounded training levels ($D^i_{kl}(\pi_\infty|\pi)$). The IAPE method has the most concentrated distribution out of all the methods, while the base method has the most disperse one. This observation is supported by Figure 5.c where we see the distribution of the per-training-instance average policies, the base policy is noticeably different from the $\infty-$level policy. Figure 6 shows the same plots for test levels, here the difference in $\Delta T_{base}|R = 10$ is less significant across methods.

a)

b)

c)

Figure 5: Results on training instances. a) Time-to-reward difference on successful training instances w.r.t the base policy. b) KL-divergence between the time-averaged policy per-training-instance of each method and the $\infty$-level method. c) Distribution of the per-training-instance average policies

a)

b)

c)

Figure 6: Results on 500 Test instances. a) Time-to-reward difference on successful test instances w.r.t the base policy. b) KL-divergence between the time-averaged policy per-test-instance of each method and the $\infty$-level method. c) Distribution of the per-test-instance average policies