[Reviews · NeurIPS 2020]

Review 1

Summary and Contributions: The author aims to improve the generalization ability of an agent to unseen (partially observable) game levels after been trained on a set of game levels which possess the same goal. To this end, the author proposes the instance agnostic policy ensemble IAPE method (with theoretical analysis) for predicting optimal actions across unseen game levels. The authors response has clarified some of my questions. However, I would like to see the writing/formalizations in the paper to be improved quite a bit to give a more clear perspective of the proposed work.

Strengths: *The formalization of gaming levels (instances) as training episodes is an interesting idea that is relevant to researchers working on generalization of RL agents. *The theoretical formalism for the transition function and value estimates for instances-based training (though some important details seem to be missing) is a valid contribution.

Weaknesses: The paper lacks many intricate details that prevents the reader to judge the novelty and full contribution of the work. After reading the rebuttal, an overview of the proposed solution and the problem setting would be of much help to the readers. *What is the markov process considered here?. Is the entire game (with all levels) considered as a POMDP? I see sentences such as “Line 62: environment is considered as a markov process”. How is the generalization problem being modelled? *Eqn 1: How is the reward function formalized? Here, the transition function is a joint distribution of the rewards and states? Usually these are two different complex functions that are modelled separately. *Remark 1 (line 112) The notion that belief state is a sufficient statistic representing the history of actions and observations and that “the approximation is always an underestimate of the value function” [Kaelbling et al. ‘98] is already proven in literature. *Def 4.1, Does a trajectory function create multiple interaction episodes, i..e, how many episodes can an instance have? Lets say many episodes are generated based on one single trajectory function, do they differ based on number of timesteps (t) or by the initial action a_0? *How are the trajectories generated in Eqn 8 (by sampling from the transition and observation distribution?). Further Fig 1b, looks similar to policy tree but since the edges represent the actions (instead of observations) it is better to have shown different action sequences. Further, how can you determine the state s_0, s_1 … if you consider a partially observable environment? *Eqn 9 says that the transition function for the overall game including all levels is the expected value of the transition function for the individual levels. How do you guarantee this ? *In Eqn 9., Assuming the trajectory function \matcal{T}^i generated multiple episodes, how can the history H_t be represented using the expectation over \matcal{T}^i_t? ? Please clearly mention what \delta and \matcal{T}(a0:t|at) means? *The markov property of the transition function has been discussed in length in the paper. How important is this transition dynamics for a partially observable environement is questionable. In a partially observable scenario, the action-observation history play a more significatn role than explicit transitions between states. The discussion about the dynamics of the policy updates with respect to the observation function is not discussed in detailed. *Def 4.2 and Lemma 3, can be seen as an extension of traditional POMDP value function and previous results [Kaelbling et al. ‘98] *In section 5, when beliefs are constructed from subset of instances, do you assume that all instance have the same set of actions, observations and the transition is formulated for the set of instances in the group?. Other notations that are left unexplained: *What is \MathCal(B) and \textbf{H} in Eqn 6, 7 *What does K mean ? does it represent the parameters of the observation distribution in Eqn 8? *What is the difference between a_(t-1) and a^i_(t-1)? In Eqn 8? *Does H_t in Eqn 9 represent one single episode generated using the trajectory function? Why is it called “full history” *T^I vs T

Correctness: Seems to be valid, provided all my assumptions (which are not explained in the paper in detail and listed as comments above) are true. Again, the experimental setup can be explained more detailed. e.g., "speed-run" policies

Clarity: No, the main contribution of the paper is difficult to capture. Several explanations seem to be vague and many notations are not explained.

Relation to Prior Work: No, some related POMDP literature (e.g., beleifs, POMDP value functions, approximations, etc) seem to be missing.

Reproducibility: Yes

Additional Feedback:


Review 2

Summary and Contributions: The paper considers the problem of generalization in RL. The paper identifies the issue of policies overfitting to the POMDP realizations (called instances) provided for training, and proposes a solution (IAPE) that leverages an ensemble of policies, each allowed to potentially overfit to a different subset of training instances. The policies are evaluated in CoinRun, where overfitting policies can be identified as doing speedruns (effective for training levels, but often ineffective in new levels). The paper shows evidence of this correspondence, and shows that IAPE generalizes better, speedruns less and produces policies that are closer to an ideal policy. The paper has interesting contributions, but the theoretical claims need to be clarified. I think the merits of this paper outweigh its issues. I would like to see the contributions and ideas of this paper presented in NeurIPS, but some of the issues on the paper would need to be adequately addressed.

Strengths: The paper has a great starting point, clearly isolating & motivating an issue, demonstrating that it happens, proposing a solution, and measuring its ability to solve both the issue and the original problem affected by the issue. It is definitely interesting work that investigates generalization in RL and work that in my opinion would be interesting to build upon.

Weaknesses: The formalization and theoretical justification of the issue need to be improved. I had difficulties parsing the notation, some definitions were missing and, overall, while the issue was clear from the outset, I found it difficult to understand that issue from the theory part of the paper.

Correctness: As mentioned, I found it difficult to understand the theory section, so the theoretical claims sound sensible and I could find varying amounts of support for them. I think the empirical claims are adequately stated and supported. There is a good mix of looking at the quantity of interest (generalization performance) and other quantities that help us diagnose whether the issue at hand is being solved.

Clarity: As with the other sections, the paper is clear except for the message in the theory section.

Relation to Prior Work: I find the discussion adequate. There is more work on belief representations that could be mentioned (see references [1-4] in the comments).

Reproducibility: Yes

Additional Feedback: Thank you for the interesting work. I have a few comments that I think could help improve the presentation. I suggest minimizing the use of beliefs in the presentation. Beliefs are a great for approaching POMDPs, but in the paper one cannot say that the belief state representation is really capturing a belief. Unless there is some attempt to demonstrate that as in [1-4], referring to representations as encoding beliefs has the risk of ascribing properties to the representation that are not there. At the end of the day, the representations are only histories processed by an RNN, and "b(H)" in Section 4 reads just as well this way as a belief. Also note that whether the processed histories are beliefs plays no role in Section 4. The important part is to what extent they capture useful information about the histories. I found Lemma 3 a bit hard to unpack. Part of the issue is that the notation on the argument of the maximizer is vague. Granted, the statements being made are not easy to express formally. My understanding of Lemma 3 is that once we are learning on a specific set of instances, the function that processes histories can ignore information that is relevant for generalization, and expose information that is beneficial for improved performance on the set of instances. Where this message mixes with the setup in the rest of the paper is that there is a common multi-instance representation that we still hope will generalize and the instance-specific implicit representation expressed by the fact that policies are allowed to overfit. The paper says about Lemma 3: "the belief function [...] will tend to overspecialize" but later "[our setup] encourages inevitable specialization to occur at the policy level". From what I understood in Lemma 3, there will be inevitable specialization happening on the representation level _and_ inevitable specialization happening at the policy level. The specialization on the policy level can be mitigated with the ensemble, but the specialization on the representation level is not discussed. Also, you could replace "inevitable" with eventual. Apart from clarifying the formal statements in Lemma 3, I would suggest some more adjustments to improve the clarity of Section 4. 1) The use of conditioning throughout was confusing to me. It's sometimes clear that the conditioning A|B means that A depends on B, but the meaning of E_{A|B}[A] is hard to parse. Is this E[A|B], or E[A] where A depends on B. For E_{A|B}[C], is it E[C], E[C|A], or E[C|A, B] (the latter two reading something like the expectation that integrates out C where the distribution underlying C is given by A, which depends on B). 2) Articulate some of the definitions in plain English. This will make the definitions in eqs. (9) easier to parse. 3) Establish the instance set more clearly. The set of instances is central to a lot that happens in the section, so it should be introduced carefully. Are there multiple sets of instances (to generalize from/to)? Within a set of instances, how are instances selected? 4) Clarify the relationship between instances and transitions. The equations lead me to think that the instances are being "resampled at each step" not really "fixed at the beginning of each episode". The actual state of the POMDP is formed by the instance and the "state" s, trajectories generated from random instances can be different from trajectories generated by the "expected transition matrix" that integrates out instances. Finally, as a technicality, I would suggest clarifying that the setup in the paper is a different paradigm from the typical DeepRL setup. It is different in the sense that the agents will realistically be exposed to the same initial conditions over and over. This setting has its own advantages and challenges, and the paper builds on the advantages in order to address the challenges. Hence the importance of highlighting the difference in setting. Other remarks: The introduction is well written, but lines 52-57 seems to rely on an understanding of CoinRun to get the point across, and this understanding is only established later in the text. I had difficulties seeing how 148-163 fit with the rest of the narrative. Please clarify what compatible means. Does it mean "has non-zero probability"? Lemma 3 does not necessarily mean sub-optimality, but that we are prone to suboptimality (and that empirical observations suggest that this is happening). So Lemma 3 says overspecialization can happen, and the empirical evidence suggests that it is happening. The conclusion of Lemma 4 can also be rephrased in the spirit of the paragraph above: Decreasing the dependence of a policy on the instance set tightens the generalization error. In this specific case you may be able to make a lower-bound statement as well, if the techniques from https://stanford.edu/class/stats311/Lectures/lec-03.pdf can be adapted to this setting. References: [1] Neural predictive belief representations (Guo et al., 2018). [2] Particle filter networks with application to visual localization (Karkus et al., 2018). [3] Neural belief states for partially observed domains (Moreno et al., 2018). [4] Shaping belief states with generative environment models for RL (Gregor et al., 2019). Typos: [26] These settings [28] the generalization gap [33] policies that generalize [eq (13)] non matching parens (twice) [297] how their performance on


Review 3

Summary and Contributions: The paper investigates overfitting to a limited set of training levels (or instances) in POMDPs. In particular, they formalize this setting and point out, that while ideally, an agent should learn to infer the correct belief over the true state, learning on limited instances can result in an agent which instead learns to infer the belief over the training instance. The authors also propose a novel training method which can alleviate those issues.

Strengths: - Better formalization of generalization in RL is very needed - The "instance-overfitting" issue raised by the authors is an interesting observation - Experimental results of their proposed algorithm are interesting and appear to confirm their hypothesis that this type of overfitting could be a relevant issue on current benchmarks. In particular, the algorithm is evaluated not on some toy problem, but on the recently proposed Coinrun benchmark. - The authors use interesting metrics in their experiments to support their hypothesis

Weaknesses: - Evaluation on other environments, even if it's another environment from the same ProcGen benchmark (which are quite diverse) would strengthen the paper significantly. - Similarly, evaluating other, already existing regularization methods (either "old" ones like Batchnorm or Dropout or "new" ones more recently proposed specifically for RL) would have been very interesting, in particular in conjunction with their novel metrics.

Correctness: Yes. See my comment about clarity.

Clarity: Yes. There was one point which I wasn't sure about: For Lemma 2, I wonder whether it is a requirement that the policy, which is evaluated, is independent from the set of instances I on which it is evaluated. Intuitively: If the policy "knows" in advance the set I on which it is evaluated, it shouldn't be an unbiased estimate of V. In the proof: I believe this might be required for the second equality in equation (20) when the integral over actions chosen by the policy, is interchanged with the expectation over I. Could you please clarify whether this is the case? If this is indeed the case, it would be helpful to highlight this in the text as I found it confusing.

Relation to Prior Work: Yes.

Reproducibility: Yes

Additional Feedback: I think this is a valuable contribution, in particular because more formalization for generalization in RL is desperately needed. However, further experimental evaluation on more benchmarks and against other baselines could substantially improve the experimental section of the paper. [Edit for after author response:] Thank you for your response. I decided to keep my score as I believe this is a valuable contribution to the community. However, I would encourage you to further improve the clarity of the paper, as pointed out in my own, and other reviews.


Review 4

Summary and Contributions: In this paper, the authors discuss the generalization properties of RL in the context of POMDPs. In the task setting the authors refer to, a single policy is required to learn on a finite set of levels, where the levels are sampled from a distribution. The policy is later tested on an unlimited levels setting, where the agent has to be generalizable in order to perform well. The authors first propose and formalize the training levels as instances, then they show that in the setting of training on finite set of instances (reusing instances), the Markov dynamics can be changed so that maximizing expected rewards leads to the memorization of overspecialize (instance-specific speed-running) policies rather than features that can be used to generalize to unseen games. The authors propose a method to tackle the above issue. They introduce the Instance agnostic policy ensemble (IAPE). In this setting, the training instances are splitted into groups. An agent is equipped with an universal encoder, which is used to generate belief representations from all instances; however, for each group of instances, a instance-subset-specific policy function takes the belief representations as input, to estimate a group specific value function, and make decisions. A consensus policy, on the other hand, is designed as the average over all subset policies. This consensus policy is used for collecting training trajectories, as well as dealing with testing environments. To verify and evaluate their assumptions and methods, the authors conduct experiments on a set of CoinRun games. They show compared with a baseline model with only BatchNorm, additional regularizations boost agents' generalization performance, an agent equipped with IAPE gives the best test performance. Further quantitative analyses show this ensemble method behaves more close to the unbounded level policy at test time. I am glad to see the authors make efforts pushing RL research to the direction of generalization, in both aspects of training on a large set of different game instances, and to test on unseen games. The RL community has focused a long time on the setting where a policy learn and test on the same single environment, and this is the extreme case where the authors pointed out --- the policy learns overspecialize features (to my understanding, it's basically overfitting). In recent years, multiple new environments have emerged to study generalization in RL. Other than the CoinRun [1] as used in this work, there are also the line of work of TextWorld [2][3], where observations and actions are both in the form of pure text; Baby AI [4] generates grid-world environments with language instructions/objectives. In addition to the difficulties introduced by the properties of datasets (e.g., language understanding), all these work report major challenge partially because RL agents show poor generalizabilities (difficult to fit multiple environments with a single policy, difficult to generalize to unseen environments etc.). This paper present one of the reasons why prior works work poorly, and experiments suggest the idea of the ensemble policy is effective. I believe this work can benefit researchers who is working on topics related to generalization in RL. Overall I like this work, I would suggest an accept. [1] Quantifying Generalization in Reinforcement Learning, Cobbe et al., 2018 [2] TextWorld: A Learning Environment for Text-based Games, Côté et al., 2018 [3] Interactive Language Learning by Question Answering, Yuan et al., 2019 [4] BabyAI: A Platform to Study the Sample Efficiency of Grounded Language Learning, Chevalier-Boisvert et al., 2019 ================== Update: Thank the authors for addressing the comments. I have read all the other reviews and the response. I like the paper, I will keep my score.

Strengths: The idea is interesting and well supported by the paper. Experimental results suggest the effectiveness of the proposed methods. Generalization in RL is a promising and useful direction. This work can benefit a group of researchers, and can potentially encourage more research in this area

Weaknesses: Nothing major I am aware of.

Correctness: To the best of my knowledge, I do not notice anything incorrect.

Clarity: The paper is well written.

Relation to Prior Work: The discussion of prior works might be a bit too simplistic/compacted.

Reproducibility: Yes

Additional Feedback: Suggestion: The authors may consider discussing the few other environments that also aim to train and test the generallizability of agents, as mentioned in the summary and contributions section. They have different input modalities than CoinRun (e.g., both Textworld and Baby AI have language components) and moreover, some games in Textworld requires to generate commands phrases, which makes the task significantly more difficult. Maybe the authors can comment on how the idea of IAPE can help on some other aspects of generalization that are absent in CoinRun.

[Author Response · NeurIPS 2020]

We thank the reviewers for their time and constructive comment. We appreciate that reviewers notice the relevance of
the presented problem, given its complexity, we will put extra effort in the presentation to clarify the mathematical
analysis and explanations as suggested. We first provide a general description of our work to clarify some of the raised
concerns, and then address the particulars. The revised paper will address all the comments.

Our goal is to provide a formulation to describe the generalization of agents trained on a finite number of levels
(instances) like the case of CoinRun. Even though the underlying model is a POMDP, for any given instance the state-
observation-reward sequences are deterministic for a given sequence of actions. We define this as the instance-specific
deterministic trajectory function, and define how it is derived from the underlying POMDP (Eq 8). This level-specific
trajectory function mirrors the role of samples in traditional supervised learning, and a model can potentially memorize
this sample without generalizing to new samples from the same distribution; yielding policies that are perfectly tuned to
the level, we draw a parallel between this phenomenon and the practice of speedrunning, where human players attempt
to complete a particular video-game level as fast as possible, the techniques used for this are often level-specific and
exploit particularities of the level that are not common throughout the rest of the environment.

Lemmas 1 and 2 show that this instance sampling procedure is internally consistent with the standard POMDP
formulation. Lemma 3 notes that an agent trained over a finite level set may exploit the characteristics of these
level-specific trajectory functions even further than one trained over an unbounded level set, yielding a policy that may
exploit essentially spurious instance-specific correlations and may not generalize to unseen instances. This is aggravated
for POMDP's because the states are unobserved and we cannot force the agent to act only based on the posterior state
distribution given past observations, rewards and actions, since these also carry information on the particular instance
the agent is acting on. This is emphasized by Lemma 4, by using standard supervised learning generalization bounds to
the learned value function. We also propose (and evaluate) how to mitigate the lack of generalization using ensembles.
All lemmas will be further explained. All suggested references and a glossary with all variables will be added.

**R1** *What is the Markov process considered here?* The entire game is considered a POMDP, we discuss a dual
representation of the environment as the standard POMDP, or the set of all possible instances sampled according to Eq
8. Generalization is modeled as the performance difference between observed and unobserved levels.

**R1** *How is the reward function formalized?* State and reward at time $t$ in the POMDP are modelled as $s_t, r_t \sim T(r_t \mid$
$s_t, a_{t-1})T(s_t \mid s_{t-1}, a_{t-1})$, we will explicitly state this in the revised version.

**R1** *Remark 1 is already proven in literature.* Citation is provided in line 109, the remark is added for context. *How*
*many episodes can an instance have?* An instance function can generate as many episodes as there are distinct action
sequences, so for a maximum episode length of $N$ and number of actions $A$ there could be as many as $A^N$ distinct
episodes in the trajectory function.

**R1,R2** *On trajectory generation.* At every node in the trajectory tree, and for every possible action a, the transition
samples its future transition from $O(o_t \mid s_t, k)T(r_t \mid s_t, a_t)T(s_t \mid a_{t-1}, s_{t-1})$. The states are latent to the environment
and unobserved by the agent. Eq 8 describes how trajectory functions are created, not how the agent perceives them
(e.g., if the agent tries the same action sequence twice on the same instance, it will get the same result, like in Eq 9).

**R1** *About Eq 9 Expectation across all levels.* Follows from Eq 8 by construction on the unbounded instance set, does not
necessarily hold for finite levels. This is better shown in Lemma 2. *Histories.* from the agent's perspective (with access
to states), the transition matrix of the next state, observation and reward is the average transition over all instances $i$
such that the transition function along the current action sequence matches the current history, $\delta$ is a Kronecker delta,
non-stochastic distribution, $\tau^i(a_{0:t} \mid a_t)$ should read $\tau^i(a_{0:t})$, the trajectory of instance $i$ along action sequence $a_{0:t}$.

**R2** *Within a set of instances, how are instances selected?* Uniformly at random at the start of the episode. *I suggest*
*minimizing the use of beliefs in the presentation* We will clarify that we are not ascribing any properties to the encoding
of the agent, these are merely latent states of our policy. The use of belief was meant to address the capabilities of
"optimal" agents.

**R2** *On Lemma 3.* Your reading of the lemma is correct, and agree on ensembles only addressing specialization at the
policy level, specialization at the representation level is addressed only via $\ell_2$ regularization and could be improved.

**R2, R3, R4** *Evaluation to other envs* We evaluated the method over three of the ProcGen environments (chaser, plunder
and dodgeball), IAPE outperforms the baseline with $\ell_2$ and batchnorm by $8\%$ to $12\%$ on total test-time episode reward.
Extended comparisons will be added.

**R3** *Lemma 2* Yes, the policy is independent of the instance set for the reasons you described.

**R1,R2** *Notations* $\mathcal{B}$: set of possible beliefs (e.g., $\mathbb{R}^b$). **H**: entropy, $K$: set of observation styles (e.g., background or
agent sprites), no influence on state, action, reward dynamics. $H_t$ in Eq 9 is a single episode from the trajectory function,
termed "full" history because it includes states, as well as observations. $T^I$ vs. $T$, $T^I$: transition matrix over instance
set $I$, $T$: transition matrix of full POMDP. Expectation notation: Will be cleaned up; $E_A[B \mid C]$ is "expectation of B
w.r.t. distribution A given C". Compatible means non-zero probability.

[Meta-Review · NeurIPS 2020]

The paper addresses the problem of generalization in POMDPs, and all reviewers agreed that it contains clever ideas which are well evaluated, and so it makes a good contribution. The reviewers also agreed that there are presentation problems that the authors should fix, but that these can be handled in a revision. Hence, I recommend acceptance and very strongly encourage the authors to revise the paper and improve the writing taking into account the detailed comments in the reviews.